# Skeletal Lipocalin-2 Is Associated with Iron-Related Oxidative Stress in ob/ob Mice with Sarcopenia

**DOI:** 10.3390/antiox10050758

**Published:** 2021-05-11

**Authors:** Eun Bee Choi, Jae Hun Jeong, Hye Min Jang, Yu Jeong Ahn, Kyu Hyeon Kim, Hyeong Seok An, Jong Youl Lee, Eun Ae Jeong, Jaewoong Lee, Hyun Joo Shin, Kyung Eun Kim, Gu Seob Roh

**Affiliations:** 1Department of Anatomy and Convergence Medical Science, Bio Anti-Aging Medical Research Center, Institute of Health Sciences, College of Medicine, Gyeongsang National University, Jinju 52777, Gyeongnam, Korea; dmsql2274@naver.com (E.B.C.); wogns3764@gmail.com (J.H.J.); gpals759@naver.com (H.M.J.); ahnujung@naver.com (Y.J.A.); gudtjr5287@hanmail.net (H.S.A.); jyv7874v@naver.com (J.Y.L.); jeasky44@naver.com (E.A.J.); woongs1111@gmail.com (J.L.); k4900@hanmail.net (H.J.S.); kke-jws@hanmail.net (K.E.K.); 2Department of Medicine, College of Medicine, Gyeongsang National University, Jinju 52777, Gyeongnam, Korea; kyuhyeunkim@gmail.com

**Keywords:** lipocalin-2, iron, inflammation, oxidative stress, sarcopenia, ob/ob mouse

## Abstract

Obesity and insulin resistance accelerate aging-related sarcopenia, which is associated with iron load and oxidative stress. Lipocalin-2 (LCN2) is an iron-binding protein that has been associated with skeletal muscle regeneration, but details regarding its role in obese sarcopenia remain unclear. Here, we report that elevated LCN2 levels in skeletal muscle are linked to muscle atrophy-related inflammation and oxidative stress in leptin-deficient ob/ob mice. RNA sequencing analyses indicated the *LCN2* gene expression is enhanced in skeletal muscle of ob/ob mice with sarcopenia. In addition to muscular iron accumulation in ob/ob mice, expressions of iron homeostasis-related divalent metal transporter 1, ferritin, and hepcidin proteins were increased in ob/ob mice compared to lean littermates, whereas expressions of transferrin receptor and ferroportin were reduced. Collectively, these findings demonstrate that LCN2 functions as a potent proinflammatory factor in skeletal muscle in response to obesity-related sarcopenia and is thus a therapeutic candidate target for sarcopenia treatment.

## 1. Introduction

Sarcopenia is a disease involving an aging-related decline in skeletal muscle mass, which results in reduced strength in elderly individuals [1]. Major risk factors for sarcopenia identified in recent studies include insulin resistance, obesity, and diabetes, conditions that, when present with sarcopenia, lead to accelerated progression of skeletal muscle atrophy [2,3]. With regard to reductions in skeletal muscle mass in obese individuals, leptin-deficient ob/ob and leptin receptor-deficient db/db genetic model mice exhibit lower muscle mass than lean littermates [4,5]. Although several mechanisms to explain the observed acceleration of sarcopenia in elderly individuals, have been proposed, including insulin resistance, excess inflammation, and upregulated oxidative stress, iron accumulation, and apoptosis [6], the details remain poorly understood. 

Skeletal muscle contains 10–15% of the body’s iron stores, primarily in mitochondria and myoglobin, which serves as a reservoir of essential iron required for the activity of iron-containing enzymes [7]. Given that iron is a transition metal that can shift between various oxidation states depending on physiologic conditions [8], uncontrolled iron accumulation is a major source of reactive oxygen species (ROS). Hence, changes in iron homeostasis in skeletal muscle can lead to muscle damage and decreased muscle function [9]. Some studies have reported an association between aging-related muscle atrophy and increased iron content and oxidative damage [10,11,12]. Although iron overload can lead to aberrant oxidative stress-induced muscle atrophy, iron deficiency can cause myopathies in both cardiac and skeletal muscle [13]. Our previous study demonstrated increased cardiac muscle iron levels in both ob/ob and db/db mice [14]. However, a causative association between elevated iron levels and sarcopenia in obese ob/ob mice has yet to be established. 

Lipocalin-2 (LCN2) is a secreted adipokine that transports small hydrophobic molecules, lipids, and iron [15]. Early research suggested that LCN2 functions as part of the innate immune system by inhibiting the growth of pathogenic bacteria via sequestration of iron, and more recent research indicates that LCN2 is involved in maintaining iron homeostasis [16]. A number of studies have demonstrated that LCN2 expression is upregulated in obese and/or diabetic rodents and humans [17,18]. Rebalka et al. reported that deletion of the *LCN2* gene impairs regeneration of skeletal muscle after cardiotoxic injury [19]. However, the potential role of LCN2 in regulating iron homeostasis and oxidative stress in obese sarcopenia remains poorly understood.

The aim of the present study was to clarify the role of LCN2 in inflammation, oxidative stress, and iron accumulation in relation to muscle atrophy, which has been associated with the acceleration of sarcopenia. 

## 2. Materials and Methods 

### 2.1. Animals

Male wild-type (WT; C57BL/6J-+/+) and leptin-deficient (ob/ob; C57BL/6J-ob/ob) mice were purchased from the Central Laboratory Animal Inc. (Seoul, Korea). The mice were 4 weeks of age at the time of purchase and were maintained in the animal facility of the Gyeongsang National University (GNU). WT mice were not littermates. Animal experiments were performed in accordance with the U.S. National Institutes of Health Guide for the Care and Use of Laboratory Animals. The University Animal Care Committee for Animal Research of GNU approved the study protocol (GNU-160530-M0025). Mice were housed under a 12-h light/12-h dark cycle. 

### 2.2. Measurements of Lean and Fat Mass

All mice underwent whole-body composition analysis using an EchoMRI system (Echo Medical Systems, Houston, TX, USA) to quantify body fat and lean mass.

### 2.3. Sample Preparation

Mice were anesthetized with zoletil (5 mg/kg; Virbac Laboratories, Carros, France). Blood samples were collected transcardially via the apex of the left ventricle. For mRNA and protein studies, the gastrocnemius muscle samples were collected and stored at −80 °C until analysis. For histologic analyses, mice were perfused with 4% paraformaldehyde (PFA) solution in 0.1 M phosphate-buffered saline (PBS). The gastrocnemius muscle of each mouse (*n* = 3–4 mice per group) was fixed in 4% PFA for 12 h at 4 °C. The resulting samples were embedded in paraffin and cut into 5-µm sections.

### 2.4. Iron Assay

Levels of total ferrous iron in frozen skeletal muscle samples were determined using an MAK025 iron assay kit (Sigma-Aldrich, St. Louis, MO, USA) according to the manufacturer’s protocol.

### 2.5. Histologic Analyses

Sections of deparaffinized skeletal muscle were incubated with Sirius Red solution (Sigma-Aldrich) for 1 h, washed with acidified water (0.5% acetic acid), dehydrated, and mounted under a coverslip with Permount (Sigma-Aldrich). The sections were observed under a BX51 light microscope (Olympus, Tokyo, Japan), and digital images were captured and documented. Collagen and non-collagen components stained red and orange, respectively. A total of five 200 × 200 μm^2^ fields were randomly selected and evaluated for each cross-section of gastrocnemius muscle. The sections were histologically scored by measuring the diameter of skeletal muscle myocytes. The distribution of myofiber sizes from the cross-sectional area was measured using iSolution software (IMT iSolution Inc., Vancouver, BC, Canada) by an observer unaware of the group assignments. Sections of deparaffinized skeletal muscle were incubated in a solution of Perls Prussian blue (Iron Stain kit, ab150674, Abcam, Cambridge, MA, USA) for 40 min to stain tissues for iron. After washing, the sections were incubated for 30 min at room temperature in a solution of 0.05% diaminobenzidine (DAB, Vector Laboratories, Burlingame, CA, USA) containing 1% H_2_O_2_. The sections were then dehydrated using a graded alcohol series, cleared using xylene, and mounted under a coverslip. The sections were observed under a BX51 light microscope (Olympus).

### 2.6. Next-Generation Sequencing (NGS)–Based RNA-Sequencing Analysis

C&K Genomics (C&K Genomics Inc., Seoul, Korea) performed RNA-seq analyses using skeletal muscles of 25-week-old WT and ob/ob mice (*n* = 3 mice per group). Briefly, sequencing was performed using an Illumina HiSeq2000 instrument, and quality-filtered reads were aligned to the *Mus_musculus* genome (GRCm38) of the Ensembl database. Differentially expressed genes (FDR < 0.001) were identified using the R package DESeq and then converted to official gene symbols and segregated into the following groups based on common biological properties determined by Gene Ontology (GO) analysis: biological process (BP), cellular component (CC), molecular function (MF). The top 100 upregulated genes in ob/ob mice were subjected to GO enrichment analysis using Enrichr (https://maayanlab.cloud/Enrichr, accessed on 3 December 2020) and compared to WT mice.

### 2.7. Quantitative Real-Time PCR (qRT-PCR)

Total RNA was isolated from frozen skeletal muscle samples (*n* = 5–6 mice per group) using TRIzol reagent (Invitrogen, Carlsbad, CA, USA) and reverse-transcribed using a RevertAid™ First-Strand cDNA Synthesis kit (Fermentas Inc., Hanover, MD, USA). For telomere length analysis, genomic DNA was extracted from muscle samples using a DNeasy Tissue kit (Qiagen, Hilden, Germany) according to the manufacturer’s protocol. RT-PCR was performed using an ABI Prism 7000 Sequence Detection System (Applied Biosystems, Foster City, CA, USA). PCR amplification was performed using a SYBR Green I qPCR kit (TaKaRa, Shiga, Japan) with specific primers (Appendix A). Glyceraldehyde-3-phosphate dehydrogenase (*GAPDH*) or *36b4* was used as a loading control for qRT-PCR analyses.

### 2.8. Western Blotting

Frozen skeletal muscle samples (*n* = 5–6 mice per group) were homogenized in lysis buffer (Thermo Fisher Scientific, Waltham, MA, USA) for immunoblotting of different proteins. The nuclear fraction of skeletal muscle cells was obtained using a NE-PER^®^ Nuclear and Cytoplasmic Extraction kit (Pierce, Rockford, IL, USA). Protein concentrations were determined using a Bio-Rad protein assay, and samples were stored at −80 °C until use. The primary antibodies used for Western blotting are detailed in Appendix A. GAPDH and p84 were used as immunoblotting loading controls, and proteins were visualized on all blots using enhanced chemiluminescence substrate (Pierce). The relative amount of each target protein was determined based on band intensity using the Multi-Gauge V 3.0 image analysis program (Fujifilm, Tokyo, Japan). Appendix A contains all uncropped images from Western blots.

### 2.9. Immunohistochemistry

Sections of deparaffinized skeletal muscle were incubated in a solution of 0.3% H_2_O_2_ for 30 min. After washing, sections were blocked by incubation for 1 h at room temperature in donkey serum. The sections were then incubated at 4 °C overnight with primary antibodies (Appendix A). After washing three times with 0.1 M PBS, the sections were then incubated for 1 h at room temperature with a secondary biotinylated antibody (1:200). After washing, the sections were incubated in avidin–biotin–peroxidase complex solution (Vector Laboratories) and developed using 0.05% DAB/horseradish peroxidase substrate (Vector Laboratories). The sections were then dehydrated using a graded alcohol series, cleared using xylene, mounted under a coverslip, and observed under a BX51 light microscope (Olympus). Immunohistochemical data for F4/80 and ferritin were obtained from selected images. Six fields (50 × 50 μm^2^) were randomly selected on each section using iSolution software (IMT iSolution Inc.) by an observer unaware of the group assignments.

### 2.10. Immunofluorescence Analysis

Sections of deparaffinized skeletal muscle were incubated with primary antibodies (Appendix A) for immunostaining. After washing three times with 0.1 M PBS, the sections were incubated for 1 h at room temperature with Alexa Fluor 488– or 594–conjugated donkey secondary antibody (Invitrogen). Nuclei were stained with 4′,6-diamidino-2-phenylindole (DAPI, 1:10,000; Invitrogen), and fluorescence images of the sections were captured using a BX51-DSU microscope (Olympus). Immunofluorescent intensity data for each protein were obtained from selected images. Six fields (50 × 50 μm^2^) were randomly selected on each section using iSolution software (IMT iSolution Inc.). Intensity measurements are represented as the percentage of the mean number of pixels versus the corresponding value at which the pixel of the respective intensity was present. 

### 2.11. Statistical Analysis

Statistical analyses were performed using PRISM 7.0 software (GraphPad Software Inc., San Diego, CA, USA). The significance of differences between WT and ob/ob mice was assessed using the Student’s *t*-test. All values are expressed as the mean ± SEM. A *p*-value of <0.05 was considered indicative of statistical significance.

## 3. Results

### 3.1. Obese ob/ob Mice Exhibit Sarcopenia with Increased Intramuscular Fat Content

Obesity often coexists with sarcopenia and accelerates the progression of aging-related muscle atrophy [20]. To determine the effects of obesity and aging on the development of sarcopenia, we initially examined the telomere length in skeletal muscle cells. Telomere length was markedly shorter in ob/ob mice compared with WT mice (Figure 1A). Although muscle atrophy in type 2 diabetic mouse models is less reliable, the most comprehensive mouse studies of diabetic sarcopenia have concentrated on the ob/ob mouse models [5,21]. The present study examined 25-week-old male ob/ob mice, which exhibit an obese phenotype with higher body weight and fat mass and lower lean mass than WT mice (Figure 1B,C). Based on reported elevations in levels of the lipid droplet marker perilipin 2 in sarcopenic muscle [22], we investigated the role of perilipin 2 in obese mice with reduced lean mass. Compared to WT mice, perilipin 2 expression was increased in the skeletal muscle of ob/ob mice (Figure 1D). As expected, analyses of cross-sections revealed an increased density of perilipin 2-immunopositive localization within myofibers in ob/ob mice compared to WT mice (Figure 1E). Excessive intramuscular fat deposits have been linked to muscle atrophy and impaired contractility [23]. Therefore, expression of the muscle atrophy markers muscle ring-finger protein-1 (MuRF1) and forkhead box protein O1 (FOXO1) was examined by Sirius Red staining and Western blotting. Consistent with the observed higher prevalence of fibrotic changes and reduced myofiber diameter in ob/ob mice (Figure 1F), the distribution of muscle fiber size showed that ob/ob mice had smaller muscle fibers compared to WT mice (Figure 1G). A significant increase in the expression of both MuRF1 and FOXO1 was observed in ob/ob mice compared to WT mice (Figure 1H,I). Double immunofluorescence analysis of muscle cross-sections of ob/ob mice revealed greater colocalization of MuRF1- and FOXO1-positive cells in ob/ob versus WT mice (Figure 1J). These findings suggest that sarcopenia in obese ob/ob mice is closely associated with the intramuscular accumulation of lipids and muscle atrophy.

### 3.2. Increased Inflammation and Oxidative Stress in Muscles of ob/ob Mice with Sarcopenia

Marked increases in the expression of genes encoding proinflammatory cytokines (i.e., tumor necrosis factor-*a* (*TNF-a*) and interleukin-6 (*IL-6*) mRNA) were observed in the skeletal muscle of ob/ob mice (Figure 2A,B). In addition, increased nuclear expression of nuclear factor kappa B p65 (NF-kBp65) and IL-6 proteins, as well as increased infiltration of F4/80-positive macrophages, were observed in skeletal muscle of ob/ob mice compared to WT mice (Figure 2C–E). Furthermore, Western blotting revealed significant increases in the expression of nuclear factor erythroid 2-related factor 2 (Nrf2), hemeoxigenase-1 (HO-1), and NADPH quinone oxidoreductase-1 (NQO-1), as well as inducible nitric oxide synthase (iNOS), in ob/ob mice compared to WT mice (Figure 2F–I). These findings suggest that sarcopenia develops as a result of mechanisms involving inflammation and oxidative stress in ob/ob mice.

### 3.3. Upregulated Expression of the LCN2 Gene in the Skeletal Muscle of ob/ob Mice

To further evaluate changes in the expression of genes associated with cell death processes, inflammation, and oxidative stress, we compared the differentially expressed genes (DEGs) in the skeletal muscle of WT and ob/ob mice using RNA-seq analysis (FDR < 0.001, logFC > 1.5; Appendix A). Among numerous DEGs identified (Figure 3A, Appendix A), the expression of mRNA encoding six genes (tissue inhibitor of metalloproteinase 1 [*Timp1*], low-density lipoprotein receptor-related protein 5 [*Lrp5*], hepatocyte growth factor [*HGF*], matrix metallopeptidase 3 [*MMP3*], tumor necrosis factor receptor superfamily, member 11a [*Tnfrsf1a*], and *LCN2*) was markedly upregulated in ob/ob mice compared to WT mice (Figure 3B). Notably, mRNA and protein levels of LCN2, which plays important roles in inflammation, hematopoietic cell apoptosis, iron and fatty acid transport, and metabolic homeostasis [14,24,25], were elevated 60- and 4-fold, respectively, in the skeletal muscle of ob/ob mice compared to WT mice (Figure 3C). In addition, mRNA and protein levels of the LCN2 receptor, 24 p3R, were also elevated (Figure 3D). As expected, double immunofluorescence analysis of LCN2 and 24p3R expression confirmed the colocalization of these proteins in muscle cross-sections of ob/ob mice compared to those of WT mice (Figure 3E). These results suggest that upregulation of skeletal LCN2 expression in ob/ob mice plays a critical role in the development of sarcopenia.

### 3.4. Iron Accumulation and Altered Expression of Iron-Regulating Proteins in Sarcopenic Muscles of ob/ob Mice

LCN2 has been implicated as playing roles in diverse physiologic processes, including inflammation, muscle regeneration, apoptosis, and iron uptake [19,26]. Analysis of GO terms, including BP, MF, and CC, was conducted for the top 100 upregulated genes identified based on *p*-value (Appendix A). Notably, these DEGs were found to be associated with iron homeostasis, inflammation, and neutrophil-mediated immunity (Appendix A). We, therefore, initially examined the distribution of iron in myofibers using DAB-enhanced Perls iron staining. As shown in Figure 4A, higher levels of iron localized predominantly in myofibers were observed in muscle cross-sections of ob/ob mice compared to WT mice. Iron assay results revealed a significant increase in the level of muscular ferrous iron (Fe^2+^) in ob/ob mice compared to WT mice (Figure 4B). 

Based on the above results, we then examined the expression of several iron-regulating proteins, including those involved in iron uptake (divalent metal transporter 1 (DMT1) and transferrin receptor), iron storage (ferritin), and iron export (ferroportin and hepcidin), using Western blotting and immunohistochemistry. Levels of DMT1 and transferrin receptor were significantly higher and lower, respectively, in the skeletal muscle of ob/ob mice compared to WT mice (Figure 4C,D). Western blotting also revealed an increase in the expression of ferritin in skeletal muscle of ob/ob mice, and increased intensity of ferritin-immunostained muscle fibers was noted in immunostaining of muscle sections from ob/ob mice (Figure 4E,F). Compared to WT mice, the level of ferroportin was reduced in ob/ob mice, but ob/ob mice exhibited a significant increase in the level of hepcidin (Figure 4G,H). However, there was no change in hepcidin expression in the livers of both WT and ob/ob mice (data not shown). Consistent with Western blotting results, lower ferroportin and higher hepcidin immunostaining were observed in the skeletal muscle of ob/ob mice (Figure 4I). Finally, levels of phosphorylated signaling transducer and activator of transcription 3 (STAT3), a transcription factor for the hepcidin gene, were significantly increased in ob/ob mice compared to WT mice (Figure 4J). Taken together, our findings suggest that sarcopenia may be associated with iron overload-mediated oxidative stress in ob/ob mice. 

## 4. Discussion

Given that obesity can accelerate the progression of skeletal muscle atrophy in the elderly, we sought to determine whether upregulation of LCN2 expression in sarcopenic muscles induces inflammation and oxidative stress in obese ob/ob mice. We demonstrate here for the first time that LCN2/24p3R-mediated iron accumulation is upregulated in obese mice with sarcopenia. Our data thus suggest that aberrant expression of iron-regulating proteins in obese sarcopenia contributes to inflammation and oxidative stress and that sarcopenia is aggravated by metabolic disorders such as obesity, insulin resistance, and type 2 diabetes.

In humans, older individuals often exhibit intramuscular fat accumulation (IMFA) with decreased muscle diameter, indicating a potential association between IMFA and sarcopenia [27]. Increased IMFA contributes to defective glucose uptake in skeletal muscle in type 2 diabetes patients [28]. Moreover, obesity-induced perimuscular fat accumulation accelerates muscle atrophy in db/db or aged mice [29]. In the present study, ob/ob mice exhibited a significant increase in fat mass relative to WT mice, but the lean mass/body weight ratio was significantly lower in ob/ob mice. These genetically obese mice exhibit a marked reduction in muscle mass and reduced capability of the muscles to undergo hypertrophy [21]. Consistent with our findings, upregulation of perilipin 2 was shown to be associated with the expression of the muscle atrophy-related gene, *MuRF-1*, suggesting a role for perilipin 2 in sarcopenia [23]. High perilipin 2 levels have been described in hepatic steatosis, atherosclerosis, sarcopenia, and some cancers [23,30,31]. It has been suggested that perilipin 2 plays an essential role in lipid storage in muscle by enhancing fatty acid uptake and triglyceride storage in myofibers, thereby blunting lipotoxicity-associated insulin resistance [31]. Furthermore, excessive secretion of certain cytokines (TNF-α and IL-6) by perimuscular fat cells is associated with various inflammatory conditions [29]. Expansion of intermuscular and perimuscular fat reserves is reportedly correlated with macrophage infiltration in high-fat diet (HFD)-fed mice [32]. Consistent with our histologic findings, macrophage marker F4/80 expression was reported to be upregulated in perimuscular adipose tissues in HFD-fed mice compared to control mice, indicating that the inflammation observed in obese sarcopenia is caused by infiltration of macrophages into peri-myocytes. It has been supported that the activation and secretion of inflammatory cytokines in obesity can lead to skeletal muscle loss, atrophy, and inflammatory myopathy [33]. Several researchers suggested that NF-kB activation accelerates inflammation and promotes protein degradation via activation of FOXO1, a transcription factor for the gene encoding E3 ubiquitin-conjugating enzyme [34]. MuRF1 and FOXO1 are typical markers of muscle atrophy in rodent model systems and humans, and these proteins are known to be involved in proteolytic pathways [35,36]. Collectively, these data suggest that fat accumulation due to obesity and secretion of inflammatory adipokines by skeletal muscle myocytes induce an inflammatory condition that plays an important role in the development of muscle diseases linked to muscle atrophy and declining muscle size. 

In addition to inflammation, obesity promotes oxidative stress due to excessive lipid peroxidation in mitochondria as well as elevations in glucose and lipid levels [37]. The resulting accumulation of oxidative damage contributes to skeletal muscle atrophy. Expression of antioxidant factors, such as Nrf2, HO-1, and NQO1, can lead to oxidative stress via the antioxidant and anti-inflammatory functions of these molecules [38]. Extensive evidence has also demonstrated that age-related increases in iron accumulation in skeletal muscle lead to increased oxidative stress and sarcopenia, which can be accelerated by obesity [8,12]. Obese ob/ob mice exhibit impaired immune responses and increased oxidative stress compared to lean littermates [39]. Consistent with our findings, other studies have demonstrated that aged rats exhibit greater inflammation and muscle atrophy along with increased iron content [8,10]. Interestingly, the smallest skeletal muscle fibers typically contained the highest levels of free iron in aged rats [40]. Skeletal muscle cells of aged rats exhibit significantly greater amounts of iron and damage associated with oxidized nucleic acids [10]. These data support our hypothesis that iron accumulation in obese sarcopenia is associated with muscle atrophy and oxidative stress via upregulation of the expression of iron-binding protein LCN2-mediated intramuscular iron translocation. 

A number of human disorders associated with obesity exhibit dysregulated expression of LCN2 [41]. One previous study reported that LCN2-knockout mice exhibit impaired skeletal muscle regeneration [19]. However, our previous research demonstrated that LCN2 overexpression in ob/ob mice exacerbates insulin resistance, inflammation, and oxidative stress in adipose tissue, heart, and brain [14,25]. Consistent with these data, LCN2 mRNA and protein levels were increased in the skeletal muscle of ob/ob mice in the present study. The results of microarray analyses indicated that leptin replacement suppresses the upregulated expression of *LCN2* and *FOXO1* related to inflammation and oxidative stress in skeletal muscles [42]. The authors of that study proposed that there is a relationship between obesity and sarcopenia associated with inflammatory conditions and suggested that leptin administration could be a beneficial therapeutic approach for preventing muscle loss. Although leptin administration decreased hyperglycemia and partially preserved muscle mass, ob/ob mice do not dramatically improve muscle mass [43]. In this regard, we hypothesize that LCN2, as a proinflammatory adipokine, plays a critical role in inflammation and iron dysregulation-mediated oxidative stress in ob/ob mice with sarcopenia. In particular, LCN2/24p3R-mediated iron uptake may cause iron accumulation, oxidative stress, and inflammation, ultimately resulting in obese sarcopenia. 

Changes in iron homeostasis play an important role in the development of aging-related sarcopenia, as iron accumulation and oxidative stress increase with age [8,12]. The increase in LCN2-related iron uptake in obese sarcopenia in the present study was similar to that observed in the heart of ob/ob mice [14]. This could explain the close correlation observed in the present study between excessive iron accumulation and skeletal muscle atrophy in ob/ob mice. Interestingly, higher levels of Perls’ staining iron accumulation and assay-determined iron concentrations in myofibers were generally observed in ob/ob mice compared to WT mice, respectively. Other studies have reported that the adverse effects of aging are associated with oxidation stress-induced iron accumulation [8,11]. Intracellular ferrous iron—also described as labile iron—is oxidized via the Haber–Weiss reaction, resulting in the formation of cellular ROS. Myocytes protect against iron overload-induced oxidative stress by reducing intracellular iron levels [44]. Several iron transport pathways play roles in myocyte iron loading, including the transferrin receptor pathway, DMT1, the LCN2/24p3R complex, and the iron exporter ferroportin [44]. Similar to increased ferritin levels and decreased transferrin receptor levels in aged rats compared to young rats [11,45], ferritin levels are also increased in the skeletal muscle of obese animals, possibly as a protective response to oxidative stress associated with ROS accumulation [44]. However, our previous study found decreased cardiac ferritin expression in ob/ob mice with left ventricular hypertrophy, which could be linked to severe damage to cardiomyocytes [14]. Our data, therefore, suggest that increased ferritin expression enhances the storage and stabilization of intracellular ferrous iron against muscular iron accumulation in ob/ob mice with sarcopenia.

Similar to enhanced iron uptake in senescent muscle, iron accumulation induced by deficiency or absence of the iron export protein ferroportin also contributes to iron overload [45]. Indeed, optimal ferroportin expression is essential for normal cardiac function [46]. Cardiac-specific deletion of ferroportin leads to fatal iron overload in cardiac muscle. In agreement with our other findings, a deleterious decrease in muscular ferroportin expression was observed in response to iron overload mediated by iron-uptake proteins, including LCN2/24p3R, DMT1, and ferritin. In particular, levels of both hepcidin and *p*-STAT3 were significantly increased in ob/ob mice compared to WT mice. Hepcidin inhibits the activity of ferroportin [47] and plays a role in inflammation; its expression is increased by STAT3 signaling via cytokines, such as IL-6 [48]. The increased expression of hepcidin observed in the present study was consistent with previous studies in model animals with chronic inflammation, such as obese mice [49]. The results of the present study are supported by previous research indicating that in morbidity obese women with non-alcoholic fatty liver disease, hepatic expression of hepcidin mRNA is significantly greater than in obese women with normal liver [50]. Our findings thus indicate that upregulation of hepcidin/STAT3 signaling contributes to the inhibition of ferroportin expression, which leads to excessive iron accumulation in the sarcopenic muscle of ob/ob mice. Collectively, our data suggest that disruption of muscular iron homeostasis due to an imbalance between iron uptake and export proteins results in iron overload-induced inflammation and oxidative stress.

The limitations of this study should be pointed out. Although we found that skeletal muscle LCN2 protein was elevated in obese ob/ob mice, we did not further elucidate the direct role of LCN2 in muscle atrophy-related mechanisms. Therefore, we intend to study the exact mechanism in the future using a LCN2 and leptin deleted knockout mice further. All of these limitations should be addressed in future studies.

## 5. Conclusions

In conclusion, obesity-induced sarcopenia is strongly associated with iron load, inflammation, and oxidative stress. The adverse effects of sarcopenia can be accelerated by metabolic disturbances associated with insulin resistance and obesity. In particular, LCN2-related iron uptake, storage, and export appear to play a critical role in obesity-related sarcopenia. Obese sarcopenia is associated with a dysregulation of muscle iron metabolism, which could be further investigated by examining LCN2/24p3R signaling-mediated muscle atrophy.

## Figures and Tables

**Figure 1 antioxidants-10-00758-f001:**
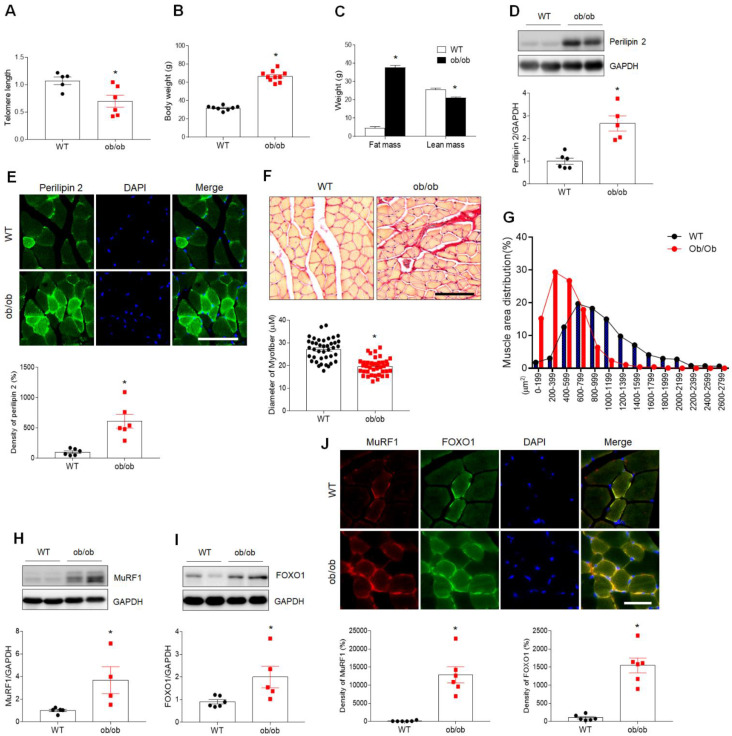
Leptin-deficient ob/ob mice exhibit characteristic sarcopenia. (**A**) Telomere length in the skeletal muscle of WT and ob/ob mice. (**B**,**C**) Body weight (**B**), and fat and lean mass (**C**) of wild-type (WT) and ob/ob mice. (**D**) Western blotting and quantitative analysis showing expression of the lipid droplet marker perilipin 2 in gastrocnemius muscles of WT and ob/ob mice. (**E**) Perilipin 2 immunofluorescence and quantitative analysis of relative fluorescent intensity in sections of skeletal muscle. (**F**) Representative images of Sirius Red staining of skeletal muscle sections showing myofibers. Mean diameter of myofibers from Sirius Red-stained cross-sections. Scale bar = 100 μm. (**G**) The distribution of myofiber sizes in skeletal muscles from WT and ob/ob mice. (**H**,**I**) Western blotting and quantitative analysis showed the expression of muscle atrophy-related markers MuRF1 (**H**) and FOXO1 (**I**) in WT and ob/ob mice. (**J**) Immunofluorescence analysis and quantitative analysis of the relative fluorescent intensity of MuRF1 and FOXO1 staining of sections of skeletal muscle. Nuclei were counterstained with DAPI. Scale bar = 25 μm. Data are shown as the mean ± SEM. * *p* < 0.05 vs. WT mice.

**Figure 2 antioxidants-10-00758-f002:**
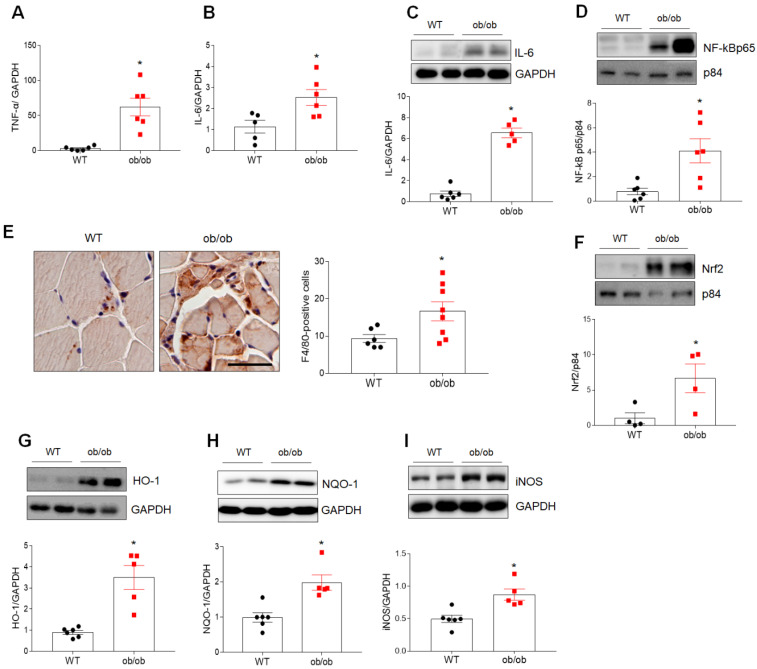
Obese sarcopenia is associated with inflammation and oxidative stress in ob/ob mice. (**A**,**B**) Quantitative RT-PCR analysis of *TNF-α* (**A**) and *IL-6* (**B**) gene expression in the skeletal muscle of wild-type (WT) and ob/ob mice. (**C**,**D**) Western blotting and quantitative analysis of IL-6 (**C**) and NF-κBp65 (**D**) expressions. (**E**) Representative images of immunostaining of F4/80 in cross-sections of skeletal muscle. The chart shows the number of F4/80-positive cells in F4/80-immunostained skeletal muscle sections. Scale bar = 50 μm. (**F**–**I**) Western blotting and quantitative analysis of Nrf2 (**F**), HO-1 (**G**), NQO-1 (**H**), and iNOS (**I**) expressions. GAPDH or p84 was used as an internal control to normalize total or nuclear protein levels, respectively. Data are shown as the mean ± SEM. * *p* < 0.05 vs. WT mice.

**Figure 3 antioxidants-10-00758-f003:**
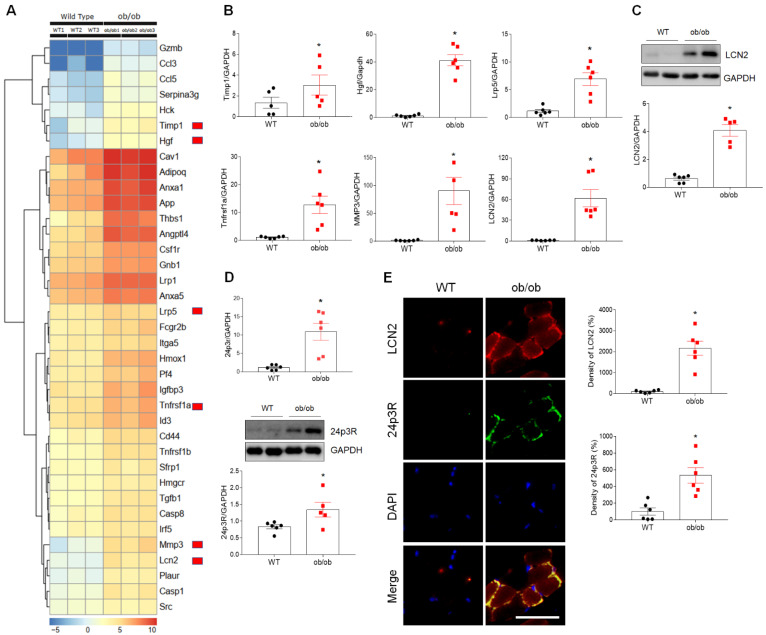
Differential expression of cell death-related genes in the skeletal muscle of wild-type (WT) and ob/ob mice. (**A**) Heat map depicting upregulated expression of cell death-related genes in ob/ob mice relative to WT mice. Red blocks indicate the six genes shown in panel B. (**B**) Quantitative RT-PCR analysis of *Timp1*, *Lrp5*, *HGF*, *MMP3*, *Tnfrsf11α*, and *LCN2* expression in skeletal muscle. (**C**) Data are shown as the mean ± SEM. * *p* < 0.05 vs. WT mice. (**C**) Western blotting and quantitative analysis of LCN2 protein (**C**) and 24p3R mRNA and protein (**D**) expression. (**E**) Representative double immunofluorescence analysis and quantitative analysis of the relative fluorescent intensity of LCN2 and 24p3R expression in cross-sections of skeletal muscle. Nuclei were counterstained with DAPI. Scale bar = 25 μm. Data are shown as the mean ± SEM. * *p* < 0.05 vs. WT mice. GAPDH was used as an internal control to normalize total protein levels.

**Figure 4 antioxidants-10-00758-f004:**
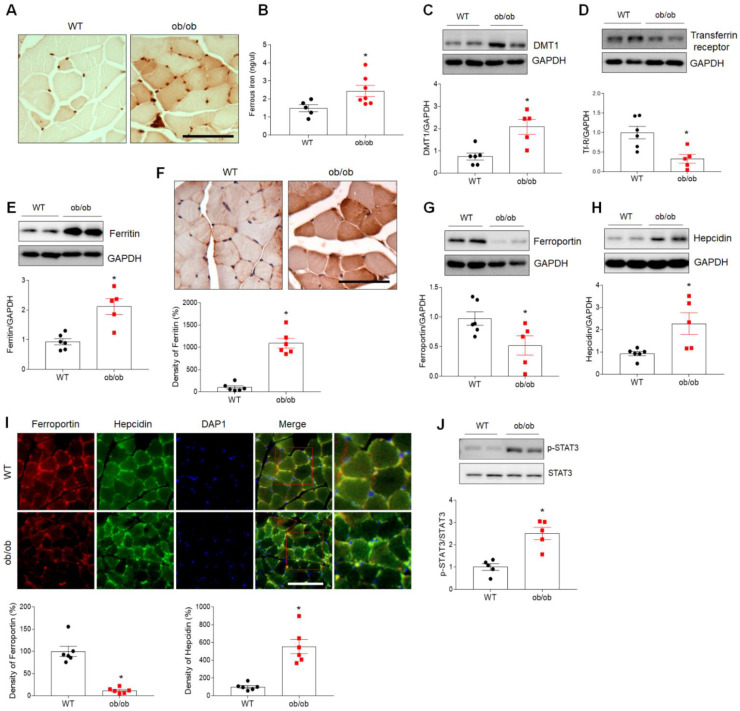
Increased expression of iron uptake-related proteins in skeletal muscle of ob/ob mice. (**A**) Histologic staining for iron using Perls’ DAB in skeletal muscle of wild-type (WT) and ob/ob mice. Scale bar = 50 µm. (**B**) Iron assay results showing levels of ferrous iron in skeletal muscle. (**C**–**E**) Western blotting and quantitative analysis of DMT1 (**C**), transferrin receptor (**D**), and ferritin (**E**) expression. GAPDH was used as an internal control to normalize total protein levels. (**F**) Representative images of immunohistochemical staining and quantitative analysis of the relative immunostained intensity of ferritin in skeletal muscle sections. Scale bar = 50 μm. (**G**,**H**) Western blotting and quantitative analysis of ferroportin (**G**) and hepcidin (**H**) expression. GAPDH was used as an internal control to normalize total protein levels. (**I**) Representative double immunofluorescence analysis and quantitative analysis of the relative fluorescent intensity of ferroportin and hepcidin expression in cross-sections of skeletal muscle. Nuclei were counterstained with DAPI. Photos on the far right are enlargements of the areas shown in red squares. Scale bar = 50 μm. (**J**) Western blotting and quantitative analysis of *p*-STAT3 and STAT3 expression. Data are shown as the mean ± SEM. * *p* < 0.05 vs. WT mice.

## Data Availability

The data presented in this study are available on request from the corresponding author.

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
