# Peer review of "Skeletal Lipocalin-2 Is Associated with Iron-Related Oxidative Stress in ob/ob Mice with Sarcopenia"

_antioxidants, 2021, doi:10.3390/antiox10050758_

Round 1

Reviewer 1 Report

This manuscript is a descriptive one, in which the authors revealed some molecular features of the muscle in ob/ob mice. In the skeletal muscle of ob/ob mice, the authors determined high levels of LCN2 levels, iron accumulation, increased expressions of iron homeostasis–related divalent metal transporter 1, ferritin, and hepcidin proteins, whereas transferrin receptor and ferroportin expressions were reduced. However, the conclusion stipulating that elevated LCN2 levels in skeletal muscle promote muscle atrophy–related inflammation and oxidative stress in leptin-deficient ob/ob mice is not supported, because no mechanism of this process is shown. Experiments in which LCN2 gene should be silenced would strengthen the conclusion of the study.   

Author Response

This manuscript is a descriptive one, in which the authors revealed some molecular features of the muscle in ob/ob mice. In the skeletal muscle of ob/ob mice, the authors determined high levels of LCN2 levels, iron accumulation, increased expressions of iron homeostasis–related divalent metal transporter 1, ferritin, and hepcidin proteins, whereas transferrin receptor and ferroportin expressions were reduced. However, the conclusion stipulating that elevated LCN2 levels in skeletal muscle promote muscle atrophy–related inflammation and oxidative stress in leptin-deficient ob/ob mice is not supported, because no mechanism of this process is shown. Experiments in which LCN2 gene should be silenced would strengthen the conclusion of the study.   

→ We thoroughly appreciate the reviewer’s critical comment. Of note, we demonstrate here for the first time that iron transporting protein LCN2-mediated iron accumulation is upregulated in skeletal muscles of obese mice with muscle atrophy. In particular, LCN2 gene has both NF-kB and STAT3 binding sites in its promotor region. So, these findings suggest that aberrant expression of iron-regulating proteins could cause inflammation (NF-kB and STAT3-mediated signaling) and oxidative stress and that sarcopenia is aggravated by metabolic disorders such as obesity and type 2 diabetes. However, as the reviewer pointed out, this study’s drawback is that the process cannot be clarified simply by increasing muscular LCN2 protein. We also suggest that further in vivo or in vitro studies using overexpression of inhibition of LCN2 are needed. Therefore we intend to further study the exact mechanism in the future using a LCN2 & leptin deleted knockout mice.

We revised some statements as follows; In abstract, “Here, we report that elevated LCN2 levels in skeletal muscle are linked to muscle atrophy–related inflammation and oxidative stress inn leptin-deficient ob/ob mice”. In conclusion, “The limitations of this study should be pointed out. Although we found that skeletal muscle LCN2 protein was elevated in obese ob/ob mice, we did not further elucidate the direct role of LCN2 in muscle atrophy-related mechanisms. Therefore, we intend to further study the exact mechanism in the future using a LCN2& leptin deleted knockout mice. All of these limitations should be addressed in future studies”.

Reviewer 2 Report

Lipocalin-2 (LCN2) is an iron-binding protein that has been associated with skeletal muscle regeneration, but details regarding its role in obese sarcopenia remain unclear. Here, the authors report that elevated LCN2 levels in skeletal muscle promote muscle atrophy-related inflammation and oxidative stress in leptin-deficient ob/ob mice. RNA sequencing analyses indicated LCN2 gene expression is enhanced in skeletal muscle of ob/ob mice with sarcopenia. In addition to muscular iron accumulation in ob/ob mice, expressions of iron homeostasis-related divalent metal transporter 1, ferritin, and hepcidin proteins were increased in ob/ob mice compared to lean littermates. Collectively, these findings demonstrate that LCN2 functions as a potent proinflammatory factor in skeletal muscle in response to obesity-related sarcopenia and is thus a therapeutic candidate target for sarcopenia treatment. Overall, this original article is highly intriguing. I think that this is acceptable for publication in Antioxidants after adding many densitometric analysis for immunofluorescence.

Major comments

  1. The author presents only microphotographs with no densitometric analysis. I think that it is very bad and poor quality. You should present the exact densitometric analysis of the percentages of positive staining for perilipin 2 (Fig. 1E), MuRF1 (Fig. 1J), FOXO1 (Fig. 1J), LCN2 (Fig. 3E), 24p3R (Fig. 3E), ferritin (Fig. 4F), Ferroportin (Fig. 4I), and Hepcldin (Fig. 4I) using several different samples (n = 5-6).

Author Response

Lipocalin-2 (LCN2) is an iron-binding protein that has been associated with skeletal muscle regeneration, but details regarding its role in obese sarcopenia remain unclear. Here, the authors report that elevated LCN2 levels in skeletal muscle promote muscle atrophy-related inflammation and oxidative stress in leptin-deficient ob/ob mice. RNA sequencing analyses indicated LCN2 gene expression is enhanced in skeletal muscle of ob/ob mice with sarcopenia. In addition to muscular iron accumulation in ob/ob mice, expressions of iron homeostasis-related divalent metal transporter 1, ferritin, and hepcidin proteins were increased in ob/ob mice compared to lean littermates. Collectively, these findings demonstrate that LCN2 functions as a potent proinflammatory factor in skeletal muscle in response to obesity-related sarcopenia and is thus a therapeutic candidate target for sarcopenia treatment. Overall, this original article is highly intriguing. I think that this is acceptable for publication in Antioxidants after adding many densitometric analysis for immunofluorescence.

Major comment

The author presents only microphotographs with no densitometric analysis. I think that it is very bad and poor quality. You should present the exact densitometric analysis of the percentages of positive staining for perilipin 2 (Fig. 1E), MuRF1 (Fig. 1J), FOXO1 (Fig. 1J), LCN2 (Fig. 3E), 24p3R (Fig. 3E), ferritin (Fig. 4F), Ferroportin (Fig. 4I), and Hepcldin (Fig. 4I) using several different samples (n = 5-6).

→ As the reviewer suggested, we measured all densitometric analysis of immunohistochemistry and immunofluorescence and added them to each figures. The methods and figure legends have been updated in revised manuscript.

Reviewer 3 Report

Thank you for the invitation to review "Skeletal lipocalin-2 is associated with iron-related oxidative stress in ob/ob mice with sarcopenia" by Choi et al. This study interrogates iron-load in the context of obesity and sarcopenia utilising a murine model. 

I have a few points for clarification:

Can the authors please state why they used only male mice in their study design?

In the abstract the authors state "Here, we report that elevated LCN2 levels in skeletal muscle promote muscle atrophy–related inflammation and oxidative stress inn leptin-deficient ob/ob mice" This is quite a bold statement and one I do not feel the authors can make, there is no mechanistic data to show that LCN is the driver of this phenotype at all. The authors show numerous singular findings, which may or may not be associated with the phenotype. To make such a claim, some intervention would be needed to k/o LCN or modify iron transport to determine this. Therefore, I feel the authors need to temper the claims in the manuscript a little to reflect this.

In the fibre CSA analysis in figure one, could the authors please present the data to show the proportional distribution of fibre CSA? I appreciate showing the mean and the individual data points, however, I think it can be presented in a more clearer way, by categorising fibre diameters on the x-axis and showing proportional distribution of the CSA.

Can the authors please provide uncrossed western blots in the supplementary material?

Lastly, the oxidative stress markers are somewhat modest and narrow in scope. Did the authors consider using measures of carbonylation, lipid peroxidation or nitrosylation - which are often used as markers when study age-related oxidate damage in skeletal muscle.

Author Response

Thank you for the invitation to review "Skeletal lipocalin-2 is associated with iron-related oxidative stress in ob/ob mice with sarcopenia" by Choi et al. This study interrogates iron-load in the context of obesity and sarcopenia utilizing a murine model. 

I have a few points for clarification:

  1. Can the authors please state why they used only male mice in their study design?

→ Thanks for your critical comment. In general, males are typically larger and more muscular than females. However, the mechanisms underlying these sex-related skeletal muscle variations are unknown, but it appears likely that they are a consequence of different sex hormonal status (estrogen). According to some studies, estrogens increase skeletal muscle force production and variations in voluntary muscle strength have been observed during human menstrual cycle (1, 2, 3). On the other hand, skeletal muscles have insulin sensitivity and regulates glucose uptake. The metabolism of obese male mice was specifically linked to insulin signaling, while the obese female mice metabolism was associated with lipid metabolism (4). Therefore, we only used male ob/ob mice.

1) Glenmark B, Nilsson M, Gao H, Gustafsson JA, Dahlman-Wright K, Westerblad H. Difference in skeletal muscle function in males vs. females: role of estrogen receptor-beta. Am J Physiol Endocrinol Metab. 2004 Dec;287(6):E1125-31.

2) Sarwar R, Niclos BB, Rutherford OM. Changes in muscle strength, relaxation rate and fatiguability during the human menstrual cycle. J Physiol. 1996 May 15;493 ( Pt 1)(Pt 1):267-72.

3) Haizlip KM, Harrison BC, Leinwand LA. Sex-based differences in skeletal muscle kinetics and fiber-type composition. Physiology (Bethesda). 2015 Jan;30(1):30-9.

4) Won EY, Yoon MK, Kim SW, et al. Gender-specific metabolomic profiling of obesity in leptin-deficient ob/ob mice by 1H NMR spectroscopy. PLoS One. 2013;8(10):e75998.

  1. In the abstract the authors state "Here, we report that elevated LCN2 levels in skeletal muscle promote muscle atrophy–related inflammation and oxidative stress inn leptin-deficient ob/ob mice" This is quite a bold statement and one I do not feel the authors can make, there is no mechanistic data to show that LCN is the driver of this phenotype at all. The authors show numerous singular findings, which may or may not be associated with the phenotype. To make such a claim, some intervention would be needed to k/o LCN or modify iron transport to determine this. Therefore, I feel the authors need to temper the claims in the manuscript a little to reflect this.

→ We thoroughly appreciate the reviewer’s critical comment. Of note, we demonstrate here for the first time that iron transporting protein LCN2-mediated iron accumulation is upregulated in skeletal muscles of obese mice with muscle atrophy. In particular, LCN2 gene has both NF-kB and STAT3 binding sites in its promotor region. So, these findings suggest that aberrant expression of iron-regulating proteins could cause inflammation (NF-kB and STAT3-mediated signaling) and oxidative stress and that sarcopenia is aggravated by metabolic disorders such as obesity and type 2 diabetes.

However, as the reviewer pointed out, this study’s drawback is that the process cannot be clarified simply by increasing muscular LCN2 protein. We also suggest that further in vivo or in vitro studies using overexpression of inhibition of LCN2 are needed. Therefore we intend to further study the exact mechanism in the future using a LCN2 & leptin deleted knockout mice.

We revised some statements as follows; In abstract, “Here, we report that elevated LCN2 levels in skeletal muscle are linked to muscle atrophy–related inflammation and oxidative stress inn leptin-deficient ob/ob mice”. In conclusion, “The limitations of this study should be pointed out. Although we found that skeletal muscle LCN2 protein was elevated in obese ob/ob mice, we did not further elucidate the direct role of LCN2 in muscle atrophy-related mechanisms. Therefore, we intend to further study the exact mechanism in the future using a LCN2& leptin deleted knockout mice. All of these limitations should be addressed in future studies”.

  1. In the fibre CSA analysis in figure one, could the authors please present the data to show the proportional distribution of fibre CSA? I appreciate showing the mean and the individual data points, however, I think it can be presented in a more clearer way, by categorising fibre diameters on the x-axis and showing proportional distribution of the CSA.

→ As the reviewer suggested, we determined the proportional distribution of the CSA and added it to figure 1G. We found that ob/ob mice had the proportion of small-sized muscle fibers compared to WT mice

  1. Can the authors please provide uncrossed western blots in the supplementary material?

→ In the supplementary figure 1, we included all uncropped images from western blots.

  1. Lastly, the oxidative stress markers are somewhat modest and narrow in scope. Did the authors consider using measures of carbonylation, lipid peroxidation or nitrosylation - which are often used as markers when study age-related oxidate damage in skeletal muscle.

→ Thanks for your critical comment. As the reviewer suggested, we additionally performed western blot analysis for 4-hydroxynonenal (4-HNE) antibody as lipid peroxidation marker. However, we found that the expression of 4-HNE in skeletal muscles of WT and ob/ob mice is not significantly different.

Round 2

Reviewer 1 Report

The authors revised the manuscript highlighting their findings and pointed out the limitation of the work. The manuscript is acceptable for publication in this form.   

Reviewer 2 Report

This revised ms is descriptively modified. I think it is possible for publication in current form. 

Reviewer 3 Report

I have no further comments.